# Transmembrane Polar Relay Drives the Allosteric Regulation for ABCG5/G8 Sterol Transporter

**DOI:** 10.3390/ijms21228747

**Published:** 2020-11-19

**Authors:** Bala M. Xavier, Aiman A. Zein, Angelica Venes, Junmei Wang, Jyh-Yeuan Lee

**Affiliations:** 1Department of Biochemistry, Microbiology and Immunology, Faculty of Medicine, University of Ottawa, Ottawa, ON K1H 8M5, Canada; bxavier@uottawa.ca (B.M.X.); azein035@uottawa.ca (A.A.Z.); avene063@uottawa.ca (A.V.); 2Biomedical Sciences Program, Department of Biology, Faculty of Science, University of Ottawa, Ottawa, ON K1N 6N5, Canada; 3Department of Pharmaceutical Sciences, School of Pharmacy, University of Pittsburgh, Pittsburgh, PA 15206, USA

**Keywords:** ABCG5, ABCG8, ATP-binding cassette transporter, cholesterol, polar relay, sitosterolemia

## Abstract

The heterodimeric ATP-binding cassette (ABC) sterol transporter, ABCG5/G8, is responsible for the biliary and transintestinal secretion of cholesterol and dietary plant sterols. Missense mutations of ABCG5/G8 can cause sitosterolemia, a loss-of-function disorder characterized by plant sterol accumulation and premature atherosclerosis. A new molecular framework was recently established by a crystal structure of human ABCG5/G8 and reveals a network of polar and charged amino acids in the core of the transmembrane domains, namely, a polar relay. In this study, we utilize genetic variants to dissect the mechanistic role of this transmembrane polar relay in controlling ABCG5/G8 function. We demonstrated a sterol-coupled ATPase activity of ABCG5/G8 by cholesteryl hemisuccinate (CHS), a relatively water-soluble cholesterol memetic, and characterized CHS-coupled ATPase activity of three loss-of-function missense variants, R543S, E146Q, and A540F, which are respectively within, in contact with, and distant from the polar relay. The results established an in vitro phenotype of the loss-of-function and missense mutations of ABCG5/G8, showing significantly impaired ATPase activity and loss of energy sufficient to weaken the signal transmission from the transmembrane domains. Our data provide a biochemical evidence underlying the importance of the polar relay and its network in regulating the catalytic activity of ABCG5/G8 sterol transporter.

## 1. Introduction

All living cells depend on the ability to translocate nutrients, metabolites, and other molecules across their membranes. One major way to achieve this is through membrane-anchored transporter proteins. The evolutionarily conserved ATP-binding cassette (ABC) transporter superfamily, for example, carries out ATP-dependent and active transport of a wide range of substances across cellular membranes, including both hydrophilic and hydrophobic molecules such as sugars, peptides, antibiotics, or cholesterol [1,2,3,4]. As a key component of cellular membranes, cholesterol constitutes ~50% of cellular lipid content; it is also the precursor of steroid hormones that modulate gene regulation and bile acids that enable nutrient absorption. Translocation of cholesterol molecules on biological membranes plays an essential role in maintaining cellular and whole-body cholesterol homeostasis. Thus, excess cholesterol needs to be eliminated from cells and tissues through either sterol acceptors in the circulation or direct excretion into the bile or the gut [5,6]. A large body of evidence indicates that ABC sterol transporters regulate cholesterol metabolism, and their defects are associated with dysregulation of whole-body cholesterol homeostasis, a major risk factor for cardiovascular diseases [7,8]. Yet, we have almost no understanding of how these transporters actually translocate cholesterol molecules and how the sterol-transport process is controlled by ATP catalysis. Given the dysregulation of cholesterol metabolism as a major risk factor for cardiovascular disease, there is a pressing need to elucidate of mechanism of these transporters in moving molecules across the cell membranes.

Recent progress in solving a heterodimeric crystal structure of human ABCG5 and ABCG8 established a new molecular framework toward such a mechanistic understanding of ABC sterol transporters. ABCG5 and ABCG8 are half-sized ABC sterol transporters and co-expressed on the apical surface of the hepatocytes along the bile ducts and the enterocytes from the intestinal brush-border membranes [9,10]. ABCG5 and ABCG8 function as obligate heterodimers (ABCG5/G8) and serve as the primary and indispensable sterol-efflux pump that effectively exports excess cholesterol, non-cholesterol sterols, and dietary plant sterols into the bile and the intestinal lumen. In mammals, most cholesterol is eliminated via its metabolism into bile acids or via biliary secretion as free cholesterol. The latter is considered as the last step of reverse cholesterol transport (RCT), where ABCG5/G8 accounts for more than 75% biliary cholesterol secretion [11,12,13,14]. Recent studies have shown that, in human subjects and animal models, ABCG5/G8 is also responsible for eliminating neutral sterols via the transintestinal cholesterol efflux (TICE), a cholesterol-lowering process independent of RCT [15]. Thus, physiologically, ABCG5/G8 plays an essential role in controlling cholesterol homeostasis in our bodies.

In general, the smallest functional unit of an ABC transporter consists of two transmembrane domains (TMD1 and TMD2) and two nucleotide-binding domains (NBD1 and NBD2), and both NBDs concertedly bind and hydrolyze ATP to provide the energy and drive substrate transport. The TMDs, on the other hand, have been shown to share low sequence similarity in the amino-acid sequences and three-dimensional structural folds, suggesting substrate-specific mechanisms for individual transporters [16]. Mechanistic analyses of ABC cholesterol transporters have largely centered on sequence requirement at the canonical ATP-binding sites [17,18,19,20], whereas little is known about the sterol–protein interaction and its relationship with ATP catalysis. Recent progress solving a crystal structure of human ABCG5/G8 revealed a unique TMD fold and several structural motifs [21]. In particular, for each subunit, a network of polar and charged amino acids is present in the core of the TMD, namely, a polar relay, whose role remains to be characterized. A triple-helical bundle is located at the transmission interface between the NBD and the TMD and consists of an elbow connecting helix, a hotspot helix (also known as an E-helix), and an intracellular loop-1 (ICL1) coupling helix. However, on the triple-helical bundle or the transmembrane polar relay, several residues have been shown to bear disease-causing missense mutations from sitosterolemia or other metabolic disorders with lipid phenotypes (Figure 1A). Notably, several disease-causing mutations are clustered in the membrane-spanning region or at the NBD–TMD interface [8,22]. This suggests the unique roles of these structural motifs in regulating the ABCG5/G8 function, yet no prior knowledge was available to explain the role of these structural motifs in the sterol-transport function.

Loss-of-function (LOF) mutations in *ABCG5* or *ABCG8* are linked to sitosterolemia, a rare autosomal recessive disease, while several other missense mutations are also associated with other lipid disorders, such as gallstone formation or elevated low-density lipoprotein (LDL) cholesterol [23,24,25,26,27,28]. At the cellular level, many of the missense mutations lead to defects in post-translational trafficking of ABCG5/G8 from the endoplasmic reticulum (ER), an abnormality commonly observed in other ABC transporters with missense mutations, e.g., ΔF508 mutation in the cystic fibrosis transmembrane conductance regulator (CFTR or ABCC7) [29,30]. However, specific missense mutants of ABCG5/G8 heterodimers have shown no defect in protein maturation [29], suggesting alternative disease-causing mechanisms. Therefore, studies of these mutants will not only show how they alter the transporter activity, but also provide mechanistic insights into the function of wild-type (WT) ABCG5/G8 sterol transporter.

Disease mutations are instrumental in studying the mechanisms of affected proteins in vitro, e.g., familial hypercholesterolemia mutations for proteins involved in low-density lipoprotein metabolism [31]. Guided by the structural framework of ABCG5/G8, we can now investigate its mechanisms using enzymological approaches with purified proteins. For this, we first need to establish at least one robust and consistent in vitro functional assay. Using ATPase activity as the functional benchmark in this study, we optimized an in vitro colorimetric ATPase assay that allows high-throughput activity assessment of detergent-purified ABCG5/G8. Using a soluble cholesterol memetic, cholesteryl hemisuccinate (CHS), we report here the CHS-stimulated ATP hydrolysis by ABCG5/G8 proteo-micelles, consisting of phospholipids, cholate, and dodecyl-maltoside (DDM), and we present an enzymatic analysis for the sterol-coupled ATPase activity on ABCG5/G8 sterol transporter. Using ATPase activity as functional readout of ABCG5/G8, we show differentially inhibition of the CHS-stimulated ATPase activity by three LOF missense mutants, two sitosterolemia mutations, and one sterol-binding mutation, where residues bearing the two disease mutations are located along the polar relay. Our data hereby demonstrate the mechanistic basis on regulating ABCG5/G8 function by the transmembrane polar relay (Figure 1B).

## 2. Results

### 2.1. CHS Stimulates ATP Hydrolysis by Wild-Type (WT) ABCG5/G8

Despite the known physiological role of ABCG5/G8 in biliary and intestinal cholesterol secretion, only indirect evidence of sterol-coupled transporter activity was detected by using steroid mimetics, such as androstane or bile acids [32,33]. In this study, we investigated a direct sterol-coupled ATPase activity by using CHS, a cholesterol mimetic that is more soluble in aqueous solution. First, to overcome low sensitivity of detecting the ABCG5/G8 ATPase activity using previous protocols, we optimized the ATPase assay for ABCG5/G8 by adopting a previous assay [34] and a colorimetric bismuth citrate-based detection approach [35]. As described and explained in Section 4, this optimized assay significantly reduces the background noise due to cloudiness by phospholipid/cholate/DDM mixtures, which improves the detecting sensitivity of liberated inorganic phosphate within the first few minutes and allows us to calculate more accurate rates of ATP hydrolysis. We show here that CHS can significantly stimulate ABCG5/G8-mediated ATP hydrolysis when co-incubated with sodium cholate (a bile acid) and *Escherichia coli* polar lipids (Figure 2). Using 5 mM ATP, the basal activity of ABCG5/G8 was calculated as 160 ± 15 nmol/min/mg (*n* = 4), similar to reported values, whereas, in the presence of CHS, the specific ATPase activity of ABCG5/G8 reached 565 ± 30 nmol/min/mg (*n* = 8), 3–4-fold higher than that in the absence of CHS (Figure 3A,B). Absence of cholate was unable to activate the ATP hydrolysis, consistent with the previous studies (data not shown) [33]. In addition, the activity was inhibited either by orthovanadate, an ATPase inhibitor [36] (Figure 3C), or by a catalytically deficient mutant ABCG5_WT_/G8_G216D_ (G8-G216D) [18], which displayed no ATP hydrolysis (Figure 3A,B). The specific activity of ATP hydrolysis by ABCG5/G8 is by far the highest in comparison with the previously reported values [33].

### 2.2. The Lipid Environments Fine-Tune ABCG5/G8 ATPase Activity

ABC transporters need to function in a phospholipid-embedded environment. However, it is unknown whether the ABCG5/G8 function is controlled by phospholipids of specific headgroups or in specific lipid compositions. Because a high concentration of bile acids is required to activate ABCG5/G8 ATPase activity, attempts to use reconstituted proteoliposomes failed due to the immediate solubilization of the reconstituted proteins. To facilitate the assessment of mutant functions, we evaluated the lipid environments to obtain the most optimal assay conditions. To study the effect of lipid conditions and phospholipid species on the ABCG5/G8 function, we analyzed the CHS-coupled ATPase activity in the presence of two polar lipid extracts under conditions of fixed concentrations of sodium cholate and CHS (see Section 4). Using *E. coli* polar lipids, we carried out an ATP concentration-dependent ATPase assay to determine the Michaelis–Menten kinetic parameters of CHS-stimulated ATP hydrolysis. We observed the maximal ATP hydrolysis by ABCG5/G8 at concentrations slightly over 2.5 mM of ATP with a V_max_ of 677.1 ± 25.6 nmol/min/mg, a K_M_(ATP) of 0.60 mM, and a k_cat_ of 1.69 s^−1^. When using bovine liver polar lipids, we observed ~3.5-fold lower catalytic rate of ATP hydrolysis and ~50% higher K_M_(ATP) (Figure 4A and Table 1). In the current study, polar lipids, cholate (bile acid), and CHS were all present in the reaction, indicating that the presence of *E. coli* polar lipids results in higher ATP association and, consequently, better stimulates ABCG5/G8 ATPase activity. When comparing the calculated values of k_cat_ and k_cat_/K_M_, we indeed observed an overall fivefold higher turnover rate in the presence of *E. coli* polar lipids than liver polar lipids (Table 1).

To determine the dependence of phospholipid headgroups, we tested the three most abundant phospholipids in either lipid extract on the ATP hydrolysis by ABCG5/G8, i.e., phosphatidylethanolamine (PE), phosphatidylcholine (PC), and phosphatidylglycerol (PG) (see Section 4). Preincubation with egg PE resulted in the highest specific activity, while the use of soy PC or egg PG only led to slightly higher ATP hydrolysis than the basal activity (Figure 4B). Interestingly, PE, the phospholipid found in both *E. coli* and liver lipids, is sufficient to stimulate ATP hydrolysis in ABCG5/G8 to almost the highest specific activity, as reported here. In the meantime, using PC or PG alone, the specific activity of ABCG5/G8 was also higher than that obtained with the liver polar lipid mixture. These results suggest phospholipid headgroups in regulating the ABCG5/G8 ATPase activity. Further investigations are necessary to pinpoint the effects of individual types of phospholipids on the sterol transporter function.

### 2.3. Missense Mutants Impair CHS-Coupled ATPase Activity of ABCG5/G8

Using the CHS-coupled ATPase activity as the functional readout, we initiated studies in the catalytic mechanism of ABCG5/G8 by exploiting the transporter’s missense mutations that undergo proper trafficking to post-ER cell membranes (ER-escaped mutants). In this study, we used *Pichia pastoris* yeast and expressed recombinant proteins of G8-G216D, a catalytically deficient mutant [18], ABCG5_E146Q_/G8_WT_ (G5-E146Q) and ABCG5_WT_/G8_R543S_ (G8-R543S), two loss-of-function/sitosterolemia missense mutants [22,37], and ABCG5_A540F_/G8_WT_ (G5-A540F), a loss-of-function mutant with putative sterol-binding defect [21] (Figure 1B and Appendix A).The purified mutants were preincubated with *E. coli* polar lipids and sodium cholate as described above. As shown in Figure 5, when compared with WT, the sitosterolemia missense mutants, G5-E146Q and G8-R543S, showed a ~80% reduction of the specific activity in CHS-coupled ATP hydrolysis (160 ± 15 nmol/min/mg and 150 ± 5 nmol/min/mg, respectively). The sterol-binding mutant G5-A540F, when compared to WT, showed a ~90% reduction of the specific activity in CHS-coupled ATP hydrolysis (90 ± 10 nmol/min/mg). Similar levels of activity reduction were also observed for non-CHS-coupled ATP hydrolysis (Appendix A). We then performed ATP concentration-dependent experiments and analyzed the Michaelis–Menten kinetics for these three mutants. For all mutants, K_M_(ATP) remained nearly the same as compared to WT, but the mutants displayed a 40–60% reduction in the catalytic rate (Table 1). This result suggests that the mutants do not alter their ability of the nucleotide association, and other molecular events contribute to the reduction of the specific ATPase activity.

The effects of CHS on ABCG5/G8 WT and mutants were further investigated by measuring the ATP hydrolysis in the CHS concentration-dependent manner at a saturated ATP concentration (5 mM here). Purified proteins were preincubated with *E. coli* polar lipids, sodium cholate, and a wide range of CHS concentrations (0.064 mM to 4.1 mM). For WT, we obtained a V_max_ of 702.9 ± 50.7 nmol/min/mg, a K_M_(CHS) of 0.79 mM, and a k_cat_ of 1.74 s^−1^ (Figure 6 and Table 2). In the presence of *E. coli* polar lipids, the catalytic rates were similar between the CHS and ATP-dependent ATPase activities, with a V_max_ of ~700 nmol/min/mg, which is about four times higher than that in the presence of liver polar lipids (Table 1 and Table 2) and more than twofold higher than the previously reported value, ~290 nmol/min/mg [33]. The catalytic rates of the mutants decreased by 70–90%, except for G5-A540F, whereas both G5-E146Q and G8-R543S displayed significantly larger K_M_(CHS), up to a twofold increase. This suggests a more profound impact of sitosterolemia mutations on the ABCG5/G8 ATPase activity through sterol–protein interaction or structural changes.

### 2.4. Missense Mutations Cause Conformational Changes at the ATP-Binding Site

To examine the relationship between structural changes of missense mutations and their impact on the ATPase activity, we performed molecular dynamics (MD) simulations for the WT and three mutants in this study. We then analyzed the MD structures to understand how the mutations could lead to different conformation around the hypothetical surrounding residues at the nucleotide-binding sites (NBS). These residues were obtained through a structural comparison between the crystal structure of ABCG5/G8 (Protein Data Bank (PDB) identifier (ID): 5DO7) and a cryo-EM structure of ABCG2 (PDB ID: 6HBU) for which two ATPs were bound in the homodimer [21,38].

To identify which residues are important for the ATP binding, we conducted MD simulations for the ABCG2 system. We calculated the ligand–residue MM-GBSA (Molecular Mechanics-Generalized Born Surface Area) free energies (ΔG_lig-res_) for the 32 surrounding residues and identified eight hotspot residues which have ΔG_lig-res_ better than −7.0 kcal/mol (Appendix A). Although those hotspots were identified for ABCG2, it is reasonable to assume they are also hotspots for ABCG5/G8 given the apparent structural and sequence similarity (only one hotspot has different amino-acid types). The root-mean-square deviation (RMSD) for the main-chain atoms was 2.60 Å, and the corresponding amino acid types of both proteins are listed in Appendix A. The detailed interactions between ATP and ABCG2 revealed by a representative MD structure are shown in Appendix A. In this study, we focused on the active nucleotide-binding site (known as NBS2) in ABCG5/G8 [21] and analyzed residues 88–103, 246–251 of ABCG5, and 210–220 and 237–245 of ABCG8. Those residues were recognized as the surrounding residues of the NBS2 in ABCG5/G8.

As shown in Figure 7, the mutations at the three sites could lead to global changes in the overall ABCG5/G8 structure, with RMSD values larger than 2.0 Å. The difference between the RMSDs of the secondary structures was smaller, probably because more obvious changes needed a longer simulation time to manifest. We were especially interested in the mutational effect on the ATP-binding site and generated RMSD vs. simulation time curves for those hypothetic surrounding residues (Appendix A). We observed that the RMSDs with and without least-square (LS) fitting were very stable for the WT, whereas, for G5-E146Q and G5-A540F, both the LS fitting and no-fitting RMSD were significantly larger. However, the G8-R543S mutation did not lead to significantly larger RMSD. This is because the distance between the mutation site and ATP binding site is greater and much longer MD simulations are required. Indeed, the RMSD had an increasing trend along the MD simulation time for G8-R543S (Appendix A). We then conducted correlation analysis using an internal program to identify possible interaction pathways between the two sites. As shown in Appendix A, the shortest path contained R543, E474, N155, V205, and L213. L213 is linked to four key residues for ATP binding. It is understandable that a perturbation at R543 needs a long simulation time to reach the ATP binding site, given that the shortest interaction path contains six residues including two ends. Overall, we observed a significant perturbation on the conformations of the putative surrounding residues due to the mutations at G5-E146Q and G5-A540F. We anticipated that the G8-R543S mutation could lead to a significant conformational change at NBS2 in much longer MD simulations.

Next, we identified representative MD conformations for all four ABCG5/G8 protein systems for comparison (Figure 8). It was observed that the hotspot residues were overlaid very well between the crystal and MD structures for the WT (Figure 8E) and R543S mutant (Figure 8H), except for R211, while, for the other two mutants, the RMSDs were significantly larger (Figure 8F,G). This observation is expected, and the reason was explained above. Interestingly, the side-chain of R211 underwent dramatic changes for all four protein systems during MD simulations. If R211 was omitted, the main-chain RMSDs became much smaller. In summary, the conformational changes from our molecular modeling could qualitatively explain why the three mutations can lead to impaired ATPase activity. Of particular note, G5-K92, the hotspot residue that has the strongest interaction with ATP, is a part of the Walker A motif at the active nucleotide-binding site and required for ABCG5/G8 functions [18,33].

## 3. Discussion

In this study, we show that CHS stimulates the ATPase activity of the human ABCG5/G8 sterol transporter to a much higher specific activity, as compared to previously reported data (Table 1 and Table 2). The much increased CHS-coupled ATPase activity indicates that ABCG5/G8 may need such a high ATP catalytic rate to achieve the sterol-transport function across the cellular membranes. CHS is a relatively water-soluble cholesterol analogue and is used to mimic cholesterol in membrane protein crystallization [21,39]. Our results showing CHS-stimulated ATPase activity suggest that the sterol molecules may have played a role in promoting an active conformation for the ATPase and/or enhancing the stability of ABCG5/G8. This idea of protein stability is supported by recent findings showing that CHS stabilizes a variety of human membrane proteins toward active conformations [40]. In the crystallographic study, >2% cholesterol was necessary to produce crystals capable of diffracting X-ray to better than 4 Å, and several sterol-like electron densities were suspected on the crystal structure of ABCG5/G8 [21]. Building upon previous work using bile acids [33] and androstane [32], our enzymatic results should come with no surprise that the WT ABCG5/G8 functionality and its active conformation are directly coupled with cholesterol analogues.

For ABCG5/G8-mediated ATP catalysis, we observed similar catalytic rates from the CHS and ATP concentration-dependent experiments, with a V_max_ of ~700 nmol/min/mg, whereas the K_M_ values were very similar to each other, K_M_(ATP) = 0.60 mM and K_M_(CHS) = 0.79 mM (Table 1 and Table 2). K_M_(ATP) and K_M_(CHS) can be used to implicate ATP and sterol association to the transporters during the ATP catalytic process, respectively. We, therefore, speculate that one ATP usage is required for sterol–protein association for one CHS (or cholesterol) molecule. Because ABCG5/G8 is believed to contain only one active NBS [18], such 1:1 stoichiometry of ATP and cholesterol for ABCG5/G8 may reflect the sterol transport rate by the single active site on this ABC transporter. An in vitro sterol-binding or transport assay, in need of development, will be necessary to directly address such a relationship. In addition to sterols, it is intriguing that PE, PC, or PG alone was sufficient to support ATPase activity of ABCG5/G8, with PE-driven activity being the highest (Figure 4). PE is the major phospholipid of the *E. coli* polar lipids, ~60%, and the second most abundant phospholipid in the bile canalicular membranes and the small intestine brush-border membranes, ~25% and ~40% respectively, of total phospholipids [41,42]. It has been shown that PE preferentially fits the headgroup-binding sites on integral membrane proteins [43]; thus, PE may be recruited as better phospholipids to support ABCG5/G8 function in the cell membranes. The approximate ratio of lipids for either *E. coli* or liver polar lipids may contribute to the apparent difference in activity, but it remains unknown how phospholipid composition regulates the transporter function. It is worth noting that specific phospholipids were shown to regulate the ATPase activity of other ABC sterol transporters, such as sphingomyelin, although the mechanistic details are not clear [19]. These individual lipids will be subjected to further examination to define the phospholipid specificity on the ABCG5/G8 ATPase activity and/or sterol-transport function.

By mapping disease-carrying residues on the apo structure of ABCG5/G8, we found that most missense variants occur within or near the structural motifs consisting of several conserved amino acids [22]. Several missense mutations (ER-trapped) prevent protein maturation from the endoplasmic reticulum (ER), but at least five mutations (ER-escaped) have been shown to undergo proper trafficking to post-ER cell membranes [29]. So far, no report has shown the impact of these ER-escaped missense mutants on ABCG5/G8 function using either in vitro or in vivo models. In this study, we used purified proteins from *Pichia pastoris* to investigate the functional activity of ABCG5/G8 in vitro and aimed to establish the mechanistic basis of ABCG5/G8 through analyzing the structure–function relationship of its loss-of-function missense mutations. The sitosterolemia missense mutants G5-E146Q and G8-R543S showed a reduction in CHS-coupled ATP hydrolysis, but retained ~20% activity as compared to WT, while the putative sterol-binding mutant G5-A540F showed further reduction to ~10% of WT ATPase activity (Figure 5 and Figure 6). With such activity reduction, the mutant proteins maintained ATPase activity similar to the basal level, as shown by WT, suggesting a remote and allosteric regulation to keep ATPase active during the reaction.

It is not uncommon that reagents such as CHS may be used as protein stabilizers for disease-causing missense variants. Here, in the absence of CHS-coupled stimulation, the mutants showed a similar level of reduced ATPase activity, arguing for a more profound effect from impaired allosteric regulation on the catalytic activity of the mutants, rather than CHS-driven stability for mutant proteins. As predicted by MD simulation, the ATP-bound homology model underwent global conformational changes upon introducing the mutations (Figure 7). These mutations, albeit relatively far away from the nucleotide-binding site, can cause significant structural rearrangement of the residues within the region that encompasses the active NBS2 (Figure 8). Such conformational changes may alter responses to the sterol–protein interaction necessary for maximal ATPase activity.

In the atomic model of ABCG5/G8 (PDB ID: 5D07), G5-E146 is located on the hotspot helix of the triple-helical bundle and in proximity to ABCG5’s polar relay, while G8-R543 is part of ABCG8’s polar relay in the core of TMD (Figure 1). Both the triple-helical bundle and the polar relay are believed to form a network of hydrogen bonding and salt bridges and play an important role in interdomain communication during the transporter function [21]. G5-E146 and G8-R543 are found in the proximity of hydrogen-bonding distance with arginine 377 of ABCG5 (G5-R377) and glutamate 503 of ABCG8 (G8-503), respectively (Figure 1B). On the basis of the ATP-dependent experiments (Figure 5 and Table 1), we obtained the changes in Gibbs free energy from WT to each mutant (ΔΔG_MUT_) as ΔΔG_E146Q_ = ~11.7 kJ/mol and ΔΔG_R543S_ = ~12.3 kJ/mol. Such energetic loss is in the range of intramolecular hydrogen-bonding potential observed on transmembrane α-helical bundles [44]. Therefore, the results support the hypothesis that the hotspot helix and the polar relay are responsible for transmitting signals between NBD and TMD. Slightly lower ΔΔG_MUT_ was observed from CHS-dependent experiments (Figure 6 and Table 2), with ΔΔG_E146Q_ = ~10.0 kcal/mol and ΔΔG_R543S_ = ~9.0 kcal/mol. This falls in the range of hydrophobic interaction and argues for weakened sterol-transporter interaction due to these disease mutations. As for the sterol-binding mutant, we obtained higher energetic loss, but similar ΔΔG_MUT_ from ATP- or CHS-dependent analysis, with ΔΔG_A540F_ = ~15.8 or ~16.1 kJ/mol, respectively. This likely indicates a strong hydrophobic interaction between sterols and the transporter, as no obvious hydrogen donors/acceptors can be found at the putative sterol-binding site on the crystal structure. In addition, G5-A540 is distant from the polar relay (>10 Å away); thus, these data suggest a remote contact by sterol molecules to control the sterol-coupled signaling, likely through the polar relay in the transmembrane domains. In the ATP concentration-dependent experiments, the K_M_ values for ATP remained almost the same (Table 1), suggesting that ATP binding was not affected by these mutants. The K_M_ values for CHS were significantly increased in the disease mutants, but not the sterol-binding mutant (Table 2), suggesting that CHS interacts with ABCG5/G8 and remotely regulates the turnover of ATP hydrolysis in either a sequential (Mode 1) or a concerted (Mode 2) pathway (Figure 9). Collectively, these results argue that a working network of hotspot helix and polar relay is essential to maintain the communication between ATPase and sterol-binding activities in ABCG5/G8, which are impaired by the loss-of-function missense mutations. As G8-R532S is the only known ER-escaped disease mutant, we will expect more insight in such polar relay-driven allosteric regulation by investigating other polar relay residues with site-directed mutagenesis.

In conclusion, these studies show that CHS stimulates ABCG5/G8 ATPase activity and may promote an active conformation for ABCG5/G8-mediated sterol transport. The enzymatic characterization of three loss-of-function missense variants provides a mechanistic basis for how the polar relay contributes to the interdomain communication for the sterol-coupled ATPase activity in ABCG5/G8 and may be directly involved in such ligand–protein interactions. Further studies will reveal more insight into these molecular events and enable sterol-lowering therapeutics to treat sitosterolemia and hypercholesterolemia.

## 4. Materials and Methods

### 4.1. Materials

*E. coli* polar lipids (Cat. #: 100600C) and bovine liver polar lipids (Cat. #: 181108C) were from Avanti Polar Lipids, Inc. (Alabaster, AL, USA). Cholesterol, cholesteryl hemisuccinate (CHS), and *n*-dodecyl β-d-maltopyranoside (DDM) were from Anatrace (Maumee, OH, USA). The nickel–nitrilotriacetic acid (Ni–NTA) agarose resin was from Qiagen Toronto (Toronto, ON, Canada), while the calmodulin (CBP) affinity resin, zeocin, and ampicillin were from Agilent (Santa Clara, CA, USA). Imidazole, ε-aminocaproic acid, sucrose, yeast extract, tryptone, peptone, yeast nitrogen base (YNB), and ammonium sulfate were obtained from Wisent Bioproducts (St-Bruno, QC, Canada). ATP disodium trihydrate, Tris-(2-carboxyethyl)-phosphine (TCEP), sodium chloride, glycerol, ethylenediaminetetraacetic acid (EDTA), ethylene glycol-bis(β-aminoethyl ether)-*N*,*N*,*N*′,*N*′-tetraacetic acid (EGTA), sodium dodecyl sulfate (SDS), Ponceau S solution, sodium azide, Bradford reagents, Tween-20, magnesium chloride, calcium chloride, and all protease inhibitors were obtained from Bioshop Canada (Burlington, ON, Canada). Biotin, sodium cholate hydrate, l-ascorbic acid, ammonium molybdate, bismuth citrate, sodium citrate, methanol, ammonium hydroxide, hydrochloric acid, and acetic acid were obtained from MilliporeSigma (Oakville, ON, Canada). Dithiothreitol (DTT), Tris base, and Tris acetate were obtained from ThermoFisher (Ottawa, ON, Canada). Clarity Western enhanced chemiluminescence (ECL) substrates, 30% acrylamide, agarose, and ammonium persulfate were obtained from Bio-Rad (Hercules, CA, USA). Restriction enzymes were obtained from New England Biolabs (NEB) (Ipswitch, MA, USA), Promega (Madison, WI, USA), and ThermoFisher (Ottawa, ON, Canada). The following media were used: yeast extract peptone dextrose (YPD), yeast extract peptone dextrose sorbitol (YPDS), minimal glycerol yeast nitrogen base (MGY), and minimal protease inhibitor buffer for yeast cell lysis (mPIB), consisting of 0.33 M sucrose, 0.3 M Tris-HCl (pH 7.5), 1 mM EDTA, 1 mM EGTA, 100 mM ε-aminocaproic acid, and ddH_2_O to a final volume of 1 L and stored at 4 °C.

### 4.2. Cloning of ABCG5/G8 Missense Mutants

The expression vectors (pLIC and pSGP18), carrying human ABCG5 (NCBI accession number NM_022436) and human ABCG8 (NCBI accession number NM_022437), were derived from pPICZB (Invitrogen) as described [33,45], pLIC-ABCG5, and pSGP18-ABCG8, respectively. A tandem array of six histidines separated by glycine (His_6_GlyHis_6_) was added to the C terminus of ABCG5, and a tag encoding a rhinovirus 3C protease site followed by a calmodulin binding peptide (CBP) was added to the C terminus of ABCG8. To generate the missense mutants in this study, we performed site-directed mutagenesis by using WT ABCG5 or ABCG8 as the templates and the following codon-optimized oligonucleotide primers (Eurofins Genomics Canada). G5-A540F: CCATTTTTGGGGTGCTTGTTGGATCTGGATTCCTCAG (forward) and GCACCCCAAAAATGGACAGCAGAGCCACTACAC (reverse); G5-E146Q: GCGCCAAACGCTGCACTACACCGCGCTGC (forward) and CAGCGTTTGGCGCACGGTGAGGCTGCTCAG (reverse); G8-R543S: GTTGCTCTATTATGGCCCTGGCCGCCGC (forward) and GCCATAATAGAGCAACAGAAGACCACCAGCCAC (reverse); G8-G216D: ACGAGCGCAGGAGAGTCAGCATTGGGGTGCAG (forward) and CTCTCCTGCGCTCGTCCCCCGACAACCCC (reverse). The polymerase chain reaction (PCR) included 1× Phusion High-Fidelity DNA Polymerase (New England Biolabs), 1× Phusion buffer, 200 mM dNTP, 2% (*v*/*v*) DMSO, 100 ng DNA templates, and 0.4 mM forward and reverse primers. Each mutant-containing plasmid was amplified by the following PCR settings: initial DNA denaturation (98 °C, 2 min), followed by 30 cycles of denaturation (98 °C, 15 s)/primer annealing (55 °C, 30 s)/DNA extension (72 °C, 3 min), and then final extension (72 °C, 20 min). Next, 5 μL of the PCR products was run on a 1% agarose gel to confirm the amplification, and 1 μL of *Dnp*I restriction enzymes (20 units) was used to digest the WT templates overnight at 37 °C. The modified plasmids were cleaned up using the ethanol acetate precipitation technique. Then, 5 μL of 3 M sodium acetate was added to each 50 μL PCR product. Next, 200 μL of 100% ethanol was added to each tube, vortexed, and left at room temperature for 10 min. At max speed in a table centrifuge for 10 min, the plasmids were pelleted; then, the supernatant was removed and washed with 75% ethanol. Residual ethanol was dried by a Speed-Vac at the maximal speed for 20 min at room temperature. The pellet was resuspended in ddH_2_O. Mutants plasmids were cloned into XL1-Blue competent *E. coli* cells by the heat-shock approach as described in the supplier’s manual (Novagen/Agilent, Santa Clara, CA, USA) and by antibiotic selection using Zeocin (Invitrogen/ThermoFisher, Ottawa, ON, Canada). Using PureYield Plasmid Midiprep kit (Promega, Madison, WI, USA), DNA preparations of selected clones were subjected to sequencing at Eurofins Genomics Canada.

### 4.3. Expression of ABCG5/G8 Missense Mutants in Pichia pastoris Yeast

Both WT and mutant plasmids (20 mg each plasmid) were linearized using *Pme*I and co-transformed into the *Pichia* strain KM71H by electroporation. Immediately, the cells were resuspended with 1 mL of ice-cold 1 M sorbitol and incubated at 30 °C for 1 h. Then, 5 mL of fresh YPD was added and incubated for 6 h at 250 rpm and 30 °C. The cells were then centrifuged at 3000× *g* for 10 min and resuspended with 200 µL of YPD. Next, 100 µL of transformants were plated on YPDS plates containing 100 (low), 500 (medium), or 1000 (high) µg/mL Zeocin to screen for successful transformation. Seven colonies were picked and grown in 10 mL of MGY medium for 24 h in sterile 50 mL tubes at 250 rpm and 30 °C. The cells were centrifuged for 10 min at 3000× *g* and then resuspended with 10 mL of minimal methanol (MM) medium. Then, 50 µL of methanol was added to the medium and once again after 12 h. The cells were harvested after 24 h incubation at 250 rpm and 30 °C, resuspended in 600 µL mPIB buffer, and transferred into a 1.5 mL Eppendorf tube. After adding 500 µL of glass beads, protease inhibitors, and 10 mM DTT, the cells were lysed using a mini-bead beater (Biospec), with 1.5 min beating and 1.5 min rest on ice for three cycles. The unbroken cells and beads were pelleted by centrifugation at 5000× *g* for 5 min at 4 °C, followed by 21,130× *g* for 5 min at 4 °C. The supernatant was collected, and the concentration of the total proteins was estimated by Bradford assay. Next, 1 μL of cell lysate and 0–10 μg BSA standards were separately added to 200 μL of Bradford reagent on a 96-well plate. Absorbance at 595 nm was used to measure the protein concentrations using a Synergy H1 Hybrid reader (BioTek/Agilent, Santa Clara, CA, USA). The cell lysates (20 or 30 μg of total proteins) were resolved by SDS–PAGE, and protein expression was analyzed by immunoblotting using monoclonal anti-RGS-His antibodies (Qiagen Toronto, Toronto, ON, Canada) to detect ABCG5 and polyclonal anti-hABCG8 antibodies (Novus Biologicals, Centennial, CO, USA) to detect ABCG8. The clones expressing the highest level for both subunits were selected and stored in 20% glycerol at −75 °C.

### 4.4. Cell Culture and Microsomal Membrane Preparation

The conditions for cell growth and WT protein induction were as previously described [21]. Briefly, cells were initially grown at 30 °C to accumulate cell mass in an Innova R43 shaker (Eppendorf) at 250 rpm for 24–48 h with the pH maintained at pH 5–6. To induce protein expression, cells were left fasting for 6–12 h, and then incubated with 0.1% (*v*/*v*) methanol for 6–12 h at 20 or 28 °C. The methanol concentration was increased to 0.5% (*v*/*v*) by adding methanol every 12 h for 48–60 h. Cell pellets were collected and resuspended in mPIB and stored at −75 °C. Approximately 45 ± 10 g of cell mass was typically obtained from 1 L of cultured cells. The frozen cells were thawed and lysed using a C3-Emulsifier (Avestin, Ottawa, ON, Canada) in mPIB in the presence of 10 mM DTT and protease inhibitors (1 μg/mL leupeptin, 1 μg/mL pepstatin A, 1 μg/mL aprotinin, and 2 mM PMSF(phenylmethylsulfonyl fluoride). The microsomal membranes were then prepared as previously described [21].

### 4.5. Purification of ABCG5/G8 and Its Mutants

Both WT and mutants were purified following a protocol described previously [21], with minor modification. Briefly, DDM-solubilized membranes were subjected to a tandem affinity column chromatography, first using Ni–NTA and then CBP. The *N*-linked glycans and the CBP tag remained on the purified heterodimers, and the CBP eluates were further purified by gel-filtration chromatography using a Superdex 200 Increase 10/300 GL column (Cytiva) on an ӒKTA Pure purification system (Cytiva, formerly GE Healthcare Life Sciences). The proteins in the peak fractions were collected and concentrated to 1–3 mg/mL for storage at −75 °C. Noticeably, the final yield for mutants was lower than WT, in a range of 400–800 µg per 6 L of cells. The expression level of the mutant proteins in the microsomes and their solubility were slightly lower than for WT. Some proteins were also lost during Ni–NTA binding and imidazole wash. The profile of the gel-filtration chromatography often showed a higher peak at the void volume than dimeric proteins. These factors collectively suggest that the mutant proteins were more prone to aggregation, thus explaining the lower yields.

### 4.6. ATPase Assay

We consistently observed a strong cloudiness in the assay solution when using previous protocols, consequently resulting in low sensitivity when detecting the ABCG5/G8 ATPase activity. Because a high concentration of bile acids is required, we reasoned that the high content of detergents, both in the assay solution and in the protein preparations, may have caused either high background upon quenching the reaction in the Malachite Green-based assay [33] or poor organic–aqueous phase separation [21]. The measurement of ATPase activity, thus, becomes inconsistent from one protein preparation to another. To overcome this issue, we first optimized the ATPase assay by adopting a colorimetric and bismuth citrate-based approach [35], which also allows high-throughput detection of the liberated inorganic phosphate by a microplate reader. The ATPase assay was performed in a 65 µL final reaction volume containing 2 mg/mL *E. coli* or liver polar lipids or designated phospholipids, 1.5% sodium cholate, 0.2% (4.11 mM) CHS, and 2 mM DTT in Buffer A (50 mM Tris/Cl pH 7.5, 100 mM NaCl, 10% glycerol, 0.1% DDM). The lipid/CHS/DTT mixture was thoroughly sonicated and preincubated with ABCG5/G8 proteins (0.3 to 1.5 µg) for 5 min at room temperature. The catalytically deficient G8-G216D was used as the negative control.

The enzymatic activity of ABCG5/G8 was initiated upon the addition of the 10× ATP cocktail (6.5 µL) and incubated at 37 °C. Aliquots (8.5 µL) were removed every 2 min and added to the prechilled quencher wells to stop the reaction. The quencher solution was made of 5% SDS in 5 mM HCl, which, together with the smaller reaction volume, contributed to a significant reduction in cloudiness for inorganic phosphate detection. Lipid mixtures were prepared at 30 mg/mL (~20 mM) in Buffer A containing 7% sodium cholate. CHS stock solution (1%, *w*/*v*) was prepared in a Buffer A and 4.5% sodium cholate, whereas 10× Mg/ATP cocktail contained 50 mM ATP, 75 mM MgCl_2_, and 100 mM NaN_3_ in a buffer containing 50 mM Tris/Cl pH 7.5. To detect the liberated inorganic phosphate, 50 µL of freshly made Solution II (142 mM ascorbic acid, 0.42 M HCl, 4.2% Solution I (10% ammonium molybdate)) was added to plate wells and left on ice for 10 min. Then, 75 µL of Solution III (88 mM bismuth citrate, 120 mM sodium citrate, 1 M HCl) was added to plate wells and placed at 37 °C for 10 min. The absorbance was measured at 695 nm using a Synergy H1 Hybrid reader (BioTek/Agilent, Santa Clara, CA, USA). For the phosphate standards, 1 M monobasic or dibasic sodium or potassium phosphate in 50 mM Tris/Cl pH 7.5 was prepared, and six standard inorganic phosphate solutions (0 μM, 12.5 μM, 25 μM, 50 μM, 100 μM, or 200 μM) were used in every experiment. The linear range of each reaction was used to calculate the initial rate of ATP hydrolysis. GraphPad Prism 8 was used to perform nonlinear regression and ordinary one-way ANOVA, with a *p*-value of ≤0.05 considered significant from at least three independent experiments. The kinetic parameters were calculated by nonlinear Michaelis–Menten curve fitting using GraphPad Prism 8.

### 4.7. Computational Methods

We studied four ABCG5/G8 protein systems including the WT and the E146Q, A540F, and R543S mutants. Each MD system consisted of one copy of ABCG5/G8 heterodimer, 320 1,2-dimyristoyl-*sn*-glycero-3-phosphocholine (DMPC) lipids, 16 cholesterols, 43,621 TIP3P [46] water molecules, and 103 Cl^−^ and 83 Na^+^ to neutralize the MD systems. AMBER ff14SB [47], Lipid14 [48], and GAFF [49] force fields were used to model proteins, DMPC lipids, and cholesterols, respectively. The residue topology of cholesterol was prepared using the Antechamber module [48]. MD simulation was performed to produce isothermal–isobaric ensembles using the pmemd.cuda program in AMBER 18 [50]. The particle mesh Ewald (PME) method [51] was used to accurately calculate the electrostatic energies with the long-ranged correction taken into account. All bonds were constrained using the SHAKE algorithm [52] in both the minimization and MD simulation stages following a computational protocol described in our previous publication [21]. Briefly, there were three stages in a series of constant-pressure and -temperature MD simulations, including the relaxation phase, the equilibrium phase, and the sampling phase. In the relaxation phase, the simulation system was heated progressively from 50 K to 250 K at steps of 50 K, and a 1 ns MD simulation was run at each temperature. In the next equilibrium phase, the system was equilibrated at 298 K, 1 bar for 10 ns. Finally, a 100 ns MD simulation was performed at 298 K, 1 bar to produce isothermal–isobaric ensemble ensembles. In total, 1000 snapshots were recorded from the last phase simulation for post-analysis using the “cpptray” module implemented in the AMBER software package. Binding free energy decomposition and correlation analysis were performed using an internal program and the detailed elsewhere [53,54].

## Figures and Tables

**Figure 1 ijms-21-08747-f001:**
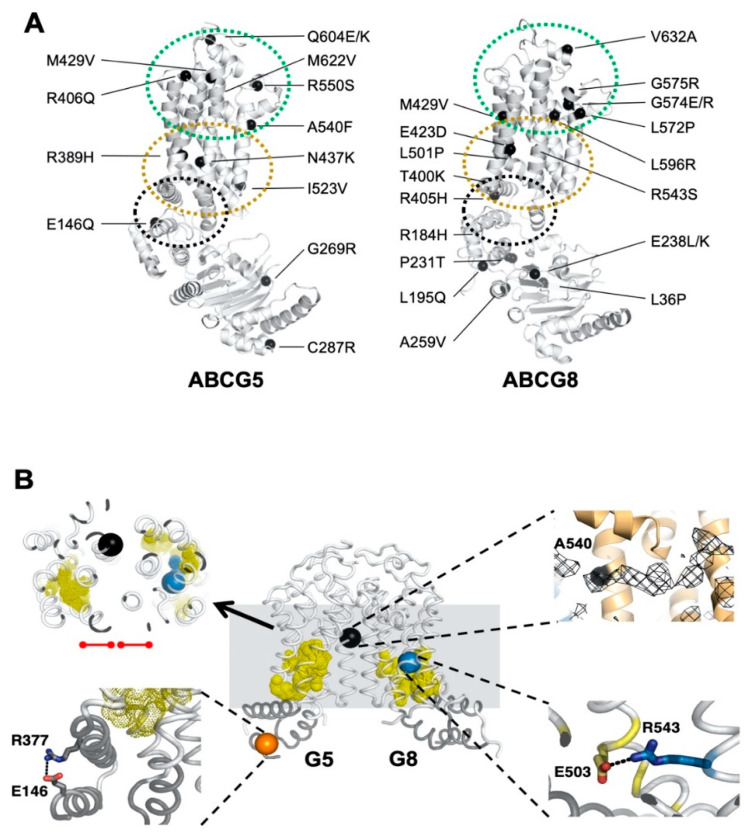
Disease-causing mutations and single-nucleotide polymorphisms (SNPs) in ATP-binding cassette (ABC) sterol transporters (ABCG5/G8). (**A**) Localization of ABCG5/G8 residues carrying missense mutations. The positions of disorder-related polymorphisms or mutations are highlighted in black spheres on the structures of ABCG5 (Protein Data Bank (PDB) identifier (ID): 5D07, chain C) and ABCG8 (PDB ID 5D07, chain D). Structural motifs are indicated in dashed ovals: triple-helical bundle (black), transmembrane domain (TMD) polar relay (yellow), and extracellular domain with re-entry helices (green). (**B**) Microenvironment of G5-E146, G5-A540, and G8-R543. (***Middle*) The transmembrane domains (white) and the triple-helical bundle (gray) are plotted in tube-styled cartoon presentation, showing the α-carbons (spheres) of G5-E146 (orange), G8-R543 (blue), and G5-A540 (black). The polar relays are plotted in dotted yellow spheres. (*Top left*) Slapped top view shows G5-A540 situated more than 10 Å away from the polar relay of either subunit (red dot-ended lines). (*Top right*) Near G5-A540 shows a cholesterol-shaped electron density (mesh) in the crystal structure of ABCG5/G8. The Fo−Fc difference electron density map was contoured at 3.0 σ. (*Bottom left*) At the triple helical bundle of ABCG5, E146 interacts with R377 through their side-chain termini in a distance of hydrogen bonding, 3.5 Å (black dashed line). (*Bottom right*) In the ABCG8 polar relay, R543 interacts E503 through their side-chain termini in a distance of hydrogen bonding, 3.1 Å (black dashed line).

**Figure 2 ijms-21-08747-f002:**
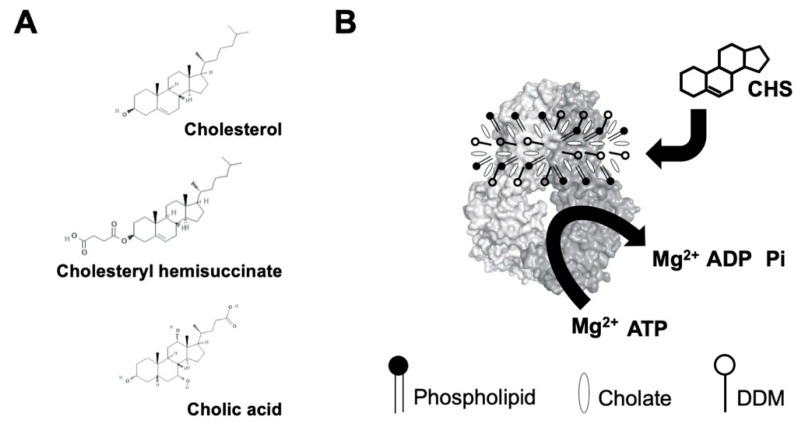
(**A**) Chemical structures of cholesterol, cholesteryl hemisuccinate (CHS), and cholic acid (cholate). Source: PubChem. (**B**) Schematic illustration of sterol-coupled ATPase activity of ABCG5/G8. Dodecyl-maltoside (DDM)-purified ABCG5/G8 (light/dark-gray surface) is preincubated with phospholipids and cholate. Addition of CHS (four-ringed steroid structure) stimulates hydrolysis of ATP to ADP and inorganic phosphate (Pi) in the presence of the divalent magnesium ions (Mg^2+^). Using the colorimetric and bismuth citrate-based assay, the liberated Pi is then captured by ammonium molybdate in the presence ascorbic acid. The color is developed upon mixing with bismuth citrate and sodium citrate, and the absorbance was measured at 695 nm. See details in Section 4.

**Figure 3 ijms-21-08747-f003:**
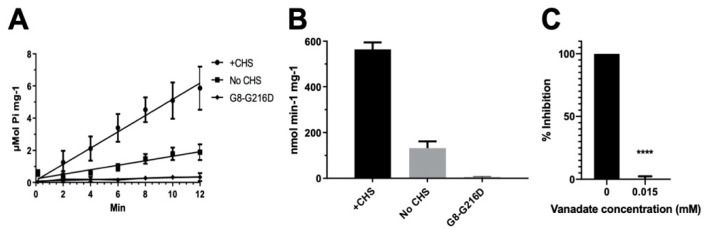
ATPase activity of ABCG5/G8. The ATP hydrolysis was used as a measure of ABCG5/G8 ATPase activity at 37 °C in conditions with 5 mM ATP and 4.1 mM CHS. The protocol is entailed in Section 4. (**A**) Data points are presented as the means ± standard deviations from 4–8 independent experiments using 2–4 independently purified proteins; where not visible, the error bars are covered by the plot symbols. A linear regression, plotted from the first 12 min, is used to calculate the specific activities. (**B**) Bar graphs show the specific activities of ATP hydrolysis by wild type (WT) in the presence and absence of CHS and the catalytically deficient mutant G8-G216D in the presence of CHS. The specific activity of WT in the absence of CHS is regarded as the basal ABCG5/G8 ATPase activity. (**C**) Bar graphs represent the percentage inhibition of ABCG5/G8 ATPase activity by 0.015 mM orthovanadate, where a *p*-value of <0.0001 (marked as ****) was obtained using ordinary one-way ANOVA (Prism 8).

**Figure 4 ijms-21-08747-f004:**
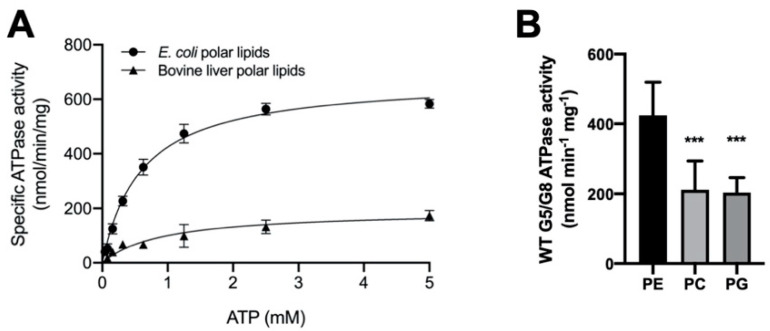
Lipid dependence of ABCG5/G8 ATPase activity. (**A**) Purified ABCG5/G8 was assayed in the presence of either *Escherichia coli* or bovine liver polar lipids, and the specific activities of ATP hydrolysis were obtained by the ATP concentration-dependent experiments (0–5 mM ATP). Both curves are fitted to the Michaelis–Menten equation (Prism 8), and, using two independently purified proteins, the means of at least three independent experiments along with standard deviations are plotted here. The kinetic parameters are listed in Table 1. (**B**) In conditions of 5 mM ATP and 4.1 mM CHS, ATP hydrolysis of purified ABCG5/G8 was assayed in the presence of egg phosphatidylethanolamine (PE), soy phosphatidylcholine (PC), or egg phosphatidylglycerol (PG), where *p*-values of 0.0006 and 0.0003 (marked as ***), respectively, were obtained using ordinary one-way ANOVA (Prism 8).

**Figure 5 ijms-21-08747-f005:**
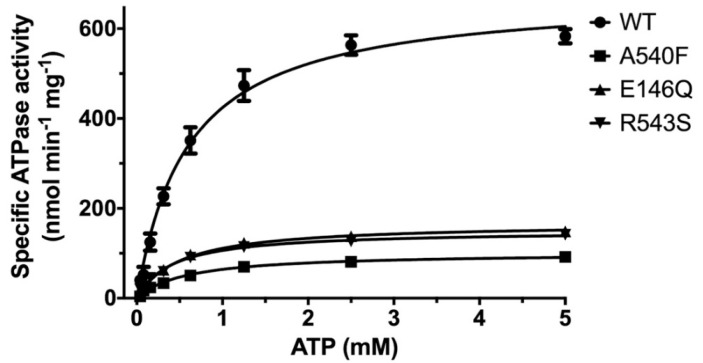
ATP dependence of ABCG5/G8 ATPase activity. Purified proteins were assayed in the presence of *E. coli* polar lipids, and the specific activities of ATP hydrolysis were obtained from the ATP concentration-dependent experiments (0–5 mM ATP). The curves are fitted to the Michaelis–Menten equation (Prism 8), and, using two-to-four independently purified proteins, the means of at least three independent experiments along with standard deviations are plotted here. The kinetic parameters are listed in Table 1.

**Figure 6 ijms-21-08747-f006:**
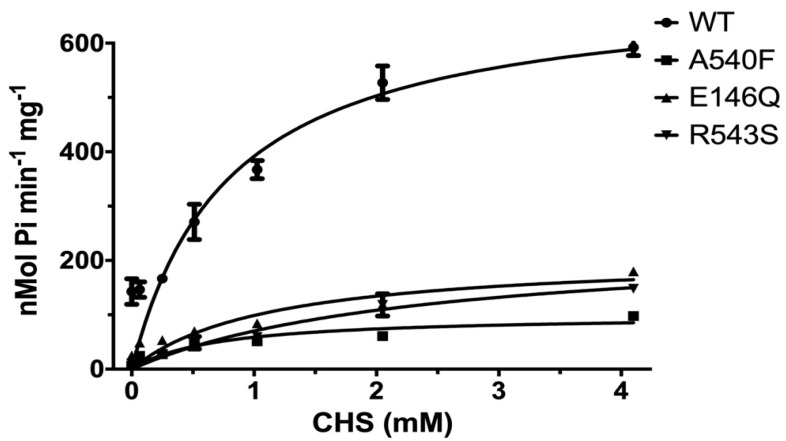
CHS dependence of ABCG5/G8 ATPase activity. Purified proteins were assayed in the presence of *E. coli* polar lipids, and the specific activities of ATP hydrolysis were obtained by the CHS concentration-dependent experiments (0–4.1 mM CHS). The curves are fitted to the Michaelis–Menten equation (Prism 8), and, using two independently purified proteins, the means of at least two independent experiments along with standard deviations are plotted here. The kinetic parameters are listed in Table 2.

**Figure 7 ijms-21-08747-f007:**
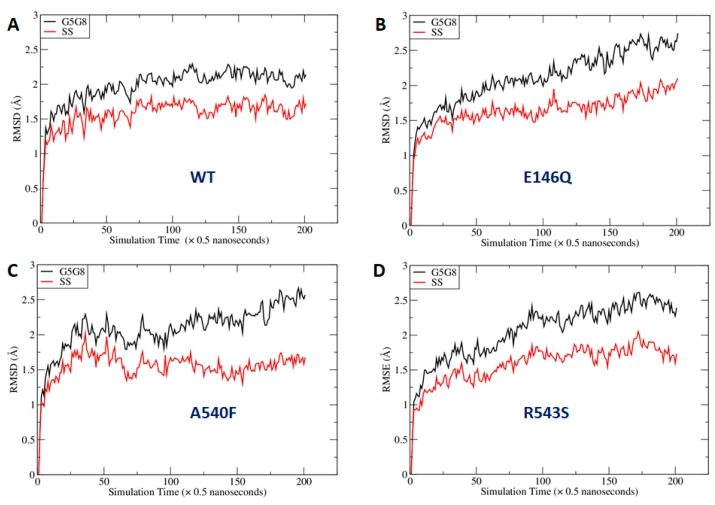
Fluctuation of root-mean-square deviations (RMSDs) along molecular dynamics (MD) simulation time course. RMSDs were calculated using the main-chain atoms of all residues (black lines) or secondary structures only (red lines): (**A**) wild type; (**B**) E146Q mutant in ABCG5; (**C**) A540F mutant in ABCG5; (**D**) R543S mutant in ABCG8. G5G8: ABCG5/G8; SS: secondary structure.

**Figure 8 ijms-21-08747-f008:**
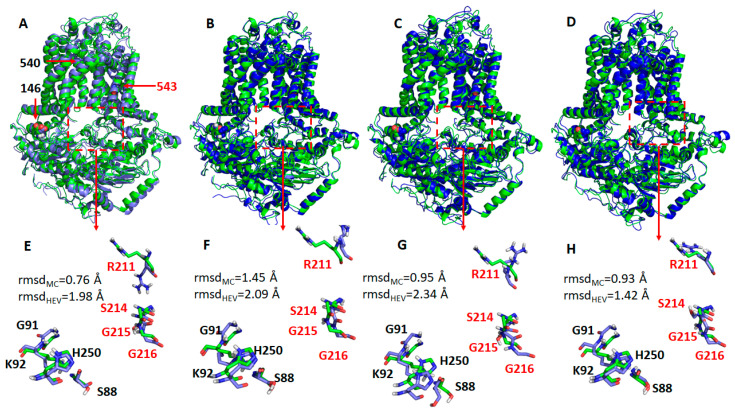
Representative structures of the WT ABCG5/G8 and its three missense mutants. The representative structures (shown as blue cartoons and bluish sticks) were aligned to the crystal structure (green cartoons, and greenish lines). The three mutation residues, E146Q, A540F, and R543S, are shown as spheres. The hypothetical surrounding residues of ATP are shown as dashed rectangles. (**A**,**E**) Wild type; (**B**,**F**) E146Q; (**C**,**G**) A540F; (**D**,**H**) R543S. G5: ABCG5; G8: ABCG8. Residues in G5 and G8 are separately colored in black and red. Root-mean-square deviations (RMSDs) for the main-chain atoms (rmsd_MC_) and all heavy atoms (rmsd_HEV_) are shown in the lower panels. If R211 is omitted from RMSD calculations, RMSDs of the main-chain atoms are 0.69, 1.30, 0.88, and 0.78 Å for WT, E146Q, A540, and R543S, respectively; the corresponding RMSDs of heavy atoms are 0.85, 1.42, 1.13, and 0.96 Å.

**Figure 9 ijms-21-08747-f009:**
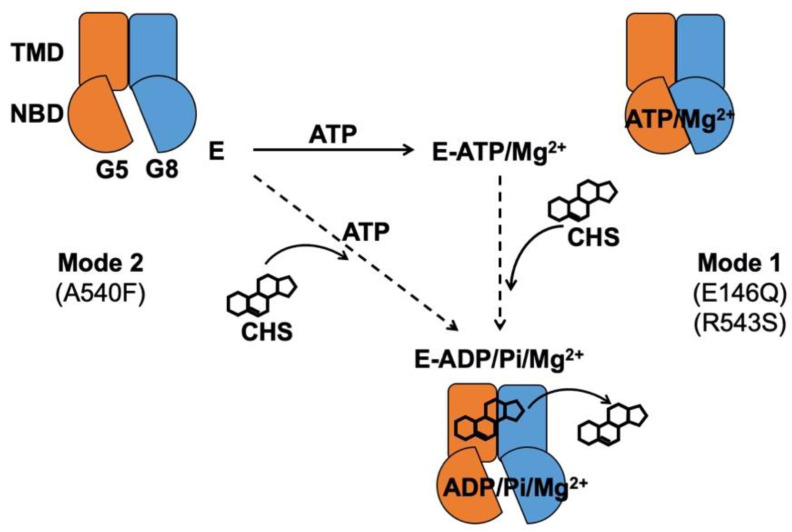
Proposed mechanism of sterol-coupled ATP catalysis by ABCG5/G8. (Mode 1) A sequential pathway is derived from experiments on the disease mutants, G5-E146Q and G8-R543S. ABCG5/G8 first recruits ATP and Mg^2+^ ions, likely causing a conformational change of the nucleotide-binding domain (NBD) for ATP binding. CHS/sterol then binds the transporter and triggers ATP hydrolysis that may result in its dissociation. (Mode 2) A concerted pathway is derived from experiments on the putative sterol-binding mutant, G5-A540F. ABCG5/G8 simultaneously recruits CHS, ATP, and Mg^2+^ ions, induces a transient conformational change of the NBD, and activates ATP hydrolysis and CHS/sterol dissociation from the transporter. G5: ABCG5; G8: ABCG8; E: ABCG5/G8 heterodimer; Pi: inorganic phosphate.

**Table 1 ijms-21-08747-t001:** Dependence of ABCG5/G8 ATPase activity on ATP.

	V_max_ ^a^ (nmol/min/mg)	K_M_ (ATP) (mM)	k_cat_ ^b^ (s^−1^)	k_cat_/K_M_ (M^−1^·s^−1^)	ΔΔG_MUT_ ^c^ (kJ/mol)	*n* ^e^
WT (liver polar lipids)	192.8 ± 17.9	0.93 ± 0.25	0.48 ± 0.04	0.52 × 10^3^	-	4
WT (*E. coli* polar lipids)	677.1 ± 25.6	0.60 ± 0.07	1.69 ± 0.06	2.8 × 10^3^	-	6
G5-E146Q ^d^	167.1 ± 0.05	0.51 ± 0.05	0.41 ± 0.00	0.82 × 10^3^	11.7	5
G8-R543S ^d^	150.7 ± 3.7	0.42 ± 0.04	0.38 ± 0.01	0.90 × 10^3^	12.3	3
G5-A540F ^d^	101.2 ± 4.2	0.58 ± 0.08	0.25 ± 0.01	0.43 × 10^3^	15.8	5

^a^ Standard errors were calculated from the fits shown in Figure 3A and Figure 5 using Prism 8 (GraphPad Software, San Diego, CA, USA). ^b^ Turnover rates, k_cat_, were calculated using the following formula: V_max_ = k_cat_ × [E], where [E] is the protein concentration of ABCG5/G8 (363.1 nM). ^c^ Differential Gibbs free energy was calculated according to the following formula: ΔΔG_MUT_ = −RTln(k_MUT_/k_WT_), where k_MUT_ is the k_cat_ of mutants, k_WT_ is the k_cat_ of WT, R = 8.314 J·mol^−1^·K^−1^ (R: gas constant), and T = 310.15 K (37 °C). ^d^ Mutants were all assayed in the presence of *E. coli* polar lipids. ^e^ Number of independent experiments.

**Table 2 ijms-21-08747-t002:** Dependence of ABCG5/G8 ATPase activity on cholesteryl hemisuccinate.

	V_max_ ^a^ (nmol/min/mg)	K_M_ (CHS) (mM)	k_cat_ ^b^ (s^−1^)	k_cat_/K_M_ (M^−1^·s^−1^)	ΔΔG_MUT_ ^c^ (kJ/mol)	*n* ^e^
WT ^d^	702.9 ± 50.7	0.79 ± 0.17	1.74 ± 0.13	2.2 × 10^3^	-	6
G5-E146Q ^d^	210.0 ± 33.2	1.13 ± 0.45	0.52 ± 0.08	0.46 × 10^3^	10.0	2
G8-R543S ^d^	237.1 ± 33.4	2.38 ± 0.67	0.59 ± 0.08	0.25 × 10^3^	9.0	2
G5-A540F ^d^	99.8 ± 11.4	0.70 ± 0.24	0.25 ± 0.03	0.36 × 10^3^	16.1	4

^a^ Standard errors were calculated from the fits shown in Figure 6 using GraphPad Prism 8. ^b^ Turnover rates, k_cat_, were calculated using the following formula: V_max_ = k_cat_ × [E], where [E] is the protein concentration of ABCG5/G8 (363.1 nM). ^c^ Differential Gibbs free energy was calculated according to the following formula: ΔΔG_MUT_ = −RTln(k_MUT_/k_WT_), where k_MUT_ is the k_cat_ of mutants, k_WT_ is the k_cat_ of WT, R = 8.314 J·mol^−1^·K^−1^ (R: gas constant), and T = 310.15 K (37 °C). ^d^ Both WT and mutants were assayed in the presence of *E. coli* polar lipids. ^e^ Number of independent experiments.

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
