# Peer review of "Transmembrane Polar Relay Drives the Allosteric Regulation for ABCG5/G8 Sterol Transporter"

_ijms, 2020, doi:10.3390/ijms21228747_

Round 1

Reviewer 1 Report

This paper presents very careful experiments on the ATPase activity of the heterodimeric ABC transporter ABCG5/G8 in the presence of various lipids and with mutations. ABCG5/G8 is needed to secrete cholesterol and other sterols, and thus mutations in this transporter result in premature atherosclerosis.

Notably, two of the mutations studied lead to disease - sitosterlomeia. The thirs mutation is in the sterol binding site. Further, one of the two disease mutations in the polar relay, while the second is nearby. First, the authors established a robust kinetic assay to study ATPase activity of ABCG5/G6 in the presence of various lipids. Kinetic data indicates the ATPase activity of ABCG5/G8 depends on the lipid composition and that sterol binding increased ATPase activity of the protein, suggesting coupling between substrate binding and ATPase activity seen for other ABC transporters. The authors perform kinetic experiments to extract the Km, Vmax, kcat of wild type and mutant proteins for ATP and CHS.

Their results show that the Km for ATP is essentially unchanged for wild type and mutant proteins, while kcat decreases. When examining the effect of CHS, the authors show that Km(CHS) increases for the polar relay mutants, but not the sterol-binding mutant, and Vmax decreases for all. The authors go further to calculate ΔΔG values betweeen WT and mutant proteins, and conduct molecular dynamics simulations in order to obtain a structural picture of how mutations alter ATPase activity.

While this paper presents a nice study showing coupling between the sterol-binding site, polar relay, and nucleotide binding site, the following revisions should be made:

  1. The authors are not reporting error values for most of their kinetic parameters - only errors in Vmax are shown.
  2. The MD simulations of wild type and mutants focus on the nucleotide binding sites, but do not mention how residues in other sites change over time with mutations. Considering the interplay between the nucleotide binding sites, polar relay, and sterol-binding site, this analysis would also be useful. Such an analysis may also explain why polar-relay mutations affect ABCG5/G8 activity with CHS, when those mutations are not in the sterol binding site.
  3. The statement that "other molecular events, e.g. sterol-transporter interaction, may have a more profound impact on ATPase activity than nucleotide association" needs to be explained better. There is no kinetic data in the manuscript for mutations in the NBS, so it isn't clear where the statement comes from. Are there data in the literature that should be referred to?
  4. There are some grammatical issues in the paper - e.g. page 8, lines 235 and 236.
  5. The Discussion reads like a results section. Much of the first 3 paragraphs can be summarized and/or moved to the Results if not already there. The authors may want to use the Discussion to expand on the structural model linking the polar relay to nucleotide binding and substrate transport.

Suggested revisions

  1. The abstract indicates that the manuscript presents an analysis on the polar relay. However, only one polar relay mutant has been studied. The authors may want to consider another mutant, perhaps one in contact with the polar relay mutant studied.

Reviewer 2 Report

This paper is an interesting combination of structural and molecular dynamics as well as biochemical experimental studies. While some of the structural data have been already published by the same authors, altogether this is an important contribution to the field of ABC transporter function and regulation. The mutation analysis is interesting and the chol-hemisuccinate activation of the G5-G8 ATPase is an especially important result.

Specific comments:

  1. Regarding sterol-activated ATPase of ABCG5-8, the statement that "direct evidence of sterol-coupled transporter activity was not available" (page 5) is not entirely valid, as earlier studies have shown such a coupling (see Muller et al, 2006).
  2. The method applied for Pi analysis is not unique to avoid lipid effects, the previous assays as described e.g. in Sarkadi et al, JBC, 1992 for such ATPase measurements are also not significantly affected by lipids.

Author Response

Response to Reviewer 2 Comments

(The revision was saved in MS Word format with the “Track Changes” option on. The page and line numbers are based on those shown under “Simple Markup” mode.)

Point 1: Regarding sterol-activated ATPase of ABCG5-8, the statement that "direct evidence of sterol-coupled transporter activity was not available" (page 5) is not entirely valid, as earlier studies have shown such a coupling (see Muller et al, 2006).

Point 2: The method applied for Pi analysis is not unique to avoid lipid effects, the previous assays as described e.g. in Sarkadi et al, JBC, 1992 for such ATPase measurements are also not significantly affected by lipids.

Response 1/2: We have made a revision to better reflect and cite previous works that have served as foundation to our studies (lines 140-141 for Point 1; lines 143-149 for Point 2). We thank the Reviewer for the suggestions and apologize for overlooking the references.

Reviewer 3 Report

The authors can measure the ATPase activity of purified ABCG5/G8 in detergent/cholate/lipid micelles and are reporting activation of the protein by cholesterol mimetic, which is a relevant result. The authors are also reporting the biochemical effect of three loss-of-function mutants, which also provides important information for a better understanding of this transporter. Furthermore, MD simulations were performed to predict the long-range structural effect of the mutations and they report some differences in the NBS2 of the mutant proteins. However, I believe that their conclusions about the molecular mechanism of the loss-of-function mutations are not fully supported by the presented data. My comments are below:

Major:

1) Page 5. For the ATPase measurements the authors referred to an optimized method that significantly reduces the background noise due to cloudiness by phospholipid/cholate/DDM mixtures. Can the authors briefly explain the reason why this method can reduce cloudiness whereas their prior methods did not?

2) Page 6: “Because a high concentration of bile acids is required to activate ABCG5/G8 ATPase activity, attempts to use reconstituted proteoliposomes failed due to the immediate solubilization of the reconstituted proteins.”  Have the authors measured the activation by CHS in absence of cholate either in detergent/lipid micelles or liposomes?

3) The A450F, E146Q, and R543S mutants showed decreased CHS-stimulated ATPase activity. What happened to the basal activity? Do these mutants have decreased basal ATPase activity, or they just failed to be activated by CHS? This information is essential to support your conclusion that “a working network of E-helix and polar relay is essential to maintain the communication between ATPase and sterol-binding activities in ABCG5/G8, which are impaired by the loss-of-function missense mutations.” (page 13). If the basal ATPase activity were not affected by the mutations, that can be a good indication of the stability of the mutant proteins, and such data would better support your conclusion. I suggest showing the ATPase activity of the mutant proteins over time, in the absence and presence of CHS (like what the authors did in Figure 3A for the WT protein). Given that the mutant proteins tend to aggregate, it is possible that the decreased ATPase activity is the result of lower protein stability and not due to failed communication between those helices and the ATPase site.

4) The A450F mutant, which is referred as a putative-sterol binding defect, showed a similar KM for CHS as the WT (0.8 mM for WT vs 0.70 mM for A450F). How is this result consistent with a sterol binding defect?

5) Page 9, Figure 6: describe the data points shown for 0mM CHS and between 100-200 specific activity. The symbols are unidentifiable. Where these data points included in the fittings?

6) Page 10: The authors state that they chose certain residues in NBS2 for analysis based on structural comparison between 5DO7 and 6HBU, but what was the actual criteria to select them? What is the importance of those residues for ATPase activity?

7) Page 11: “It is observed that three positively charged residues, K92, H49 and R50 of ABCG5 are overlaid very well between the crystal and MD structures for the WT (Figure 8E), while for the three mutants, at least one residue cannot be superimposed.” Can you specify which residues are not superimposing in each structure? That sentence seems to implicate that those three positively charged residues are all well aligned with the crystal structure only for the WT, but the other MD structures in the figure do not seem to show much difference for these residues.

8) Page 11: “Therefore, the conformational changes from our molecular modeling can qualitatively explain why the three mutations can lead to impaired ATPase activity.” I do not think the information provided is enough to explain why the mutations can lead to impaired activity. Can the authors elaborate more to justify such conclusion? For example: it is not clear how the inability to superimpose some of those residues would influence ATP binding. According to the figure 8E, the R52 residue is also more distant from the center of the ATP binding site in the WT protein and does not overlap well with the crystal structure. The orientation of K92 (in Walker A) does not seem to be very different between WT and the mutants. Which of the MD predicted changes is expected to impair ATPase activity? I understand that E146Q and A540F showed increased RMSD for NBS2 but going into specific amino acids without further explanation do not seem to really provide essential information.

9) In the figure legends the authors refer to “at least three replicates”. Were those replicates obtained from at least three independent protein purifications or are they replicates from the same purified sample? Appendix Figure 1 show 4-6 colonies and the authors mentioned they selected the clones with the higher expression of both G5 and G8, but they do not specify which ones or how many total independent protein purified samples were used for the experiments. Also, the independent number of samples is not reported for the tables.

Minor:

- Legend of Figure 1B should read “G5-E146, G5-A540, and G8-R343.” The A of alanine is missing.

- Page 8. Please add references associated to the functional effect of the mutations and for the putative-sterol binding defect.

- Figure 8. For parts B, C, and D, show spheres only for the corresponding mutation instead of the three of them.

- Legends of tables 1 and 2 refer to figure 5D, but there is no figure 5D. It also refers to the fits in figure 3A, but that figure does not have data for liver lipids or information about the mutants.

Round 2

Reviewer 3 Report

I would suggest that the authors modify the following part of the discussion: "The sitosterolemia missense mutants G5-E146Q and G8-R543S have shown a reduction of CHS-coupled ATP hydrolysis, but retained ~20% activity as compared to WT, while the putative sterol-binding mutant G5-A540F has shown further reduction to ~10% of WT ATPase activity (Figures 5 & 6). With such activity reduction, the mutant proteins maintained the ATPase activity similar to the basal level, as shown by WT, suggesting a remote and allosteric regulation to keep ATPase active during the reaction". There the authors are still comparing basal activity of the WT with CHS-estimulated activity of the mutants and that can be misleading. They must clarify in that sentence as well that the mutants displayed lower basal ATPase activity compared to the WT, as their new data showed, but the mutants still showed activation by CHS. So the presence of a single mutation did not fully impair the substrate-induced activation (allosteric communication?).